# Linking Pregnancy and Long-Term Health: The Impact of Cardiovascular Risk on Telomere Shortening in Pregnant Women

**DOI:** 10.3390/medicina59061012

**Published:** 2023-05-24

**Authors:** Simona-Alina Abu-Awwad, Marius Craina, Adrian Gluhovschi, Paula Diana Ciordas, Catalin Marian, Lioara Boscu, Elena Bernad, Mircea Iurciuc, Ahmed Abu-Awwad, Stela Iurciuc, Anca Laura Maghiari

**Affiliations:** 1Doctoral School, “Victor Babes” University of Medicine and Pharmacy, 300041 Timisoara, Romania; alina.abuawwad@umft.ro; 2“Clinic of Obstetrics and Gynecology”, “Pius Brinzeu” County Clinical Emergency Hospital, 300723 Timisoara, Romania; mariuscraina@hotmail.com (M.C.); adigluhovschi@yahoo.com (A.G.); ahm.abuawwad@umft.ro (A.A.-A.); boscu.anca@umft.ro (A.L.M.); 3Department of Obstetrics and Gynecology, Faculty of Medicine, “Victor Babes” University of Medicine and Pharmacy, 300041 Timisoara, Romania; 4Center for Laparoscopy, Laparoscopic Surgery and In Vitro Fertilization, “Victor Babes” University of Medicine and Pharmacy, 300041 Timisoara, Romania; 5Departament IV—Discipline of Biochemistry, “Victor Babes” University of Medicine and Pharmacy, 300041 Timisoara, Romania; paulamuntean22@gmail.com (P.D.C.); cmarian@umft.ro (C.M.); 6Department of Cardiology, “Victor Babes” University of Medicine and Pharmacy, 300041 Timisoara, Romania; lioara.boscu@umft.ro (L.B.); mirceaiurciuc@gmail.com (M.I.); iurciuc.stela@umft.ro (S.I.); 7Departament VI—Discipline of Outpatient Internal Medicine, Cardiovascular Prevention and Recovery, “Victor Babes” University of Medicine and Pharmacy, 300041 Timisoara, Romania; 8Department XV—Discipline of Orthopedics—Traumatology, “Victor Babes” University of Medicine and Pharmacy, 300041 Timisoara, Romania; 9Research Center University Professor Doctor Teodor Șora, “Victor Babes” University of Medicine and Pharmacy, 300041 Timisoara, Romania; 10Departament I—Discipline of Anatomy and Embryology, “Victor Babes” University of Medicine and Pharmacy, 300041 Timisoara, Romania

**Keywords:** pregnancy, pregnant woman, high risk, pathology, cardiovascular disease, telomere

## Abstract

*Background and Objectives*: Telomeres are repetitive DNA sequences located at the end of chromosomes that play a crucial role in maintaining chromosomal stability. Shortening of telomeres has been associated with an increased risk of cardiovascular disease. The aim of this study was to investigate whether the length of telomeres in pregnant women with cardiovascular risk is shorter compared to those without cardiovascular risk. *Materials and Methods*: A total of 68 participants were enrolled, including 30 pregnant women with cardiovascular risk and 38 without cardiovascular risk, who were followed-up during their pregnancy between 2020 and 2022 at the Obstetrical and Gynecology Department of the “Pius Brînzeu” Emergency County Clinical Hospital in Timişoara, Romania. All included women underwent delivery via cesarean section at the same medical institution. The telomere length was measured in each participant using quantitative Polymerase chain reaction (PCR). *Results*: The results showed that the telomere length was negatively correlated with cardiovascular risk in pregnant women, with significantly shorter telomeres observed in the cardiovascular risk group (mean telomere length = 0.3537) compared to the group without cardiovascular risk (mean telomere length = 0.5728) (*p* = 0.0458). *Conclusions*: These findings suggest that cardiovascular risk during pregnancy may be associated with accelerated telomere shortening, which could have implications for the long-term health of both the mother and the child. Further research is needed to investigate the potential mechanisms underlying this association and to identify interventions that may mitigate the negative effects of cardiovascular risk on the telomere length during pregnancy.

## 1. Introduction

Telomeres, which are repetitive nucleotide elements located at the terminal portion of both arms of most eukaryotic organisms’ and some prokaryotes’ chromosomes, function to protect the chromosomes from degradation and preserve genetic information [1]. As cells divide, the telomere length gradually shortens, resulting in telomere loss in all dividing cells, including diploid cells. Therefore, the telomere length is reduced over time. Studies have shown that telomere attrition is also linked to the ability to replicate in vitro [2]. Additionally, it can predict the lifespan.

Telomeres play an important role in the management of cellular senescence and the aging of the body. They must be protected to avoid DNA damage and to trigger DNA damage response pathways. Very short telomeres are recognized as dysfunctional telomeres. They lead to replicative cellular senescence and chromosomal instability: both being hallmarks of aging.

Telomeres have become an important area of research in understanding the relationship between aging and disease risk [3]. A shortened telomere length has been linked to an increased risk of several chronic diseases, including cardiovascular disease (CVD) [4]. Recently, there has been growing interest in exploring the relationship between telomere length and pregnancy-related CVD risk.

Pregnancy is a complex physiological process that places significant stress on the maternal cardiovascular system [5]. The changes in hormone levels and blood volume during pregnancy result in significant hemodynamic adaptations, which can have long-term consequences for the mother’s health. Pregnancy-related complications, such as gestational hypertension, preeclampsia, and gestational diabetes, have been linked to an increased risk of developing CVD later in life [6]. Telomere length has been proposed as a potential biomarker for predicting pregnancy-related CVD risk.

Telomere shortening can influence the health condition in a negative way. A lot of health issues, including aging and cancer, have been linked to telomere shortening.

The objective of this study was to examine the difference in the telomere length between pregnant women who have cardiovascular disease or associated cardiovascular risk factors and those who do not.

The current development suggests that the telomere length may serve as a potential biomarker for predicting pregnancy-related cardiovascular disease risk, but more research is needed to fully understand this relationship.

## 2. Materials and Methods

This study was performed in the Department of Obstetrics and Gynecology at the ‘Pius Brinzeu’ County Emergency Clinical Hospital from Timisoara, from 1 January 2020 to 31 December 2022.The research population and its relevant features were identified by using a population-based administrative database of patients who received outpatient care at the same clinic throughout the study period. The medical records of patients were centrally stored in a database that complied with privacy laws, and these records were obtained with the patients’ consent. The database contained information such as the patients’ demographic data, medical history, and in-hospital procedures. The baseline characteristics and procedures of all patients were recorded both in the hospital database and in paper patient records, which were reviewed by certified clinicians involved in the study. The importance of having an interconnected database that maintains patient information regardless of their location cannot be overstated. With advancements in technology and the increasing mobility of people, patients may receive medical care from different providers and in different locations. As a result, it is crucial for medical professionals to have access to the most up-to-date and accurate information about their patients. An interconnected database would allow medical professionals to easily and quickly access a patient’s medical history, test results, diagnoses, and other important information, regardless of where they were previously treated. This could lead to improved patient outcomes, as medical professionals would have a more comprehensive understanding of a patient’s health status and medical needs [7].

Sixty-eight pregnant women were included in the study, thirty-eight without cardiovascular risk or cardiovascular disease (group 1) and thirty with cardiovascular risk or cardiovascular disease (group 2).

Cardiovascular risk refers to the likelihood of developing cardiovascular disease, which includes conditions such as coronary artery disease, heart attack, stroke, and other disorders that affect the heart and blood vessels. In the context of this group of patients, cardiovascular risk refers to the presence of risk factors or pre-existing conditions that increase the likelihood of developing cardiovascular disease during or after pregnancy.

All the included women were admitted for delivery through caesarean section (C-section) at the same medical institution. A caesarean section is a surgical procedure in which a baby is delivered through an incision made in the mother’s abdomen and uterus. This method of delivery is typically used when a vaginal birth is not possible or safe for the mother or the baby.

This study enrolled participants who met the following inclusion criteria:Pregnant women in their second or third trimester.Women aged between 18 and 40 years.Women who have been diagnosed with cardiovascular disease or have at least one cardiovascular risk factor, such as hypertension, diabetes, obesity, or a family history of cardiovascular disease.Women who are willing to participate in the study and provide a blood sample for telomere length analysis.Women who are not currently taking any medications that may affect the telomere length, such as hormone therapy.Women who have not been diagnosed with any other chronic diseases that may affect the telomere length, such as cancer or autoimmune disorders.Women who have received prenatal care and have had regular check-ups throughout their pregnancy.Women who have provided informed consent to participate in the study.Pregnancy completed by caesarean section.A history of no more than two miscarriages.Negative COVID-19 history in the last year.

This study excluded participants who met the following exclusion criteria:Women who have a history of drug or alcohol abuse.Women who have a history of psychiatric disorders or mental health issues.Women who have a history of thromboembolic disease or clotting disorders.Women who have participated in other clinical trials or studies within the past 3 months.Women who have a history of adverse reactions to blood draws or phlebotomy.Women who are unable or unwilling to provide informed consent to participate in the study.Patients with infection conditions, such as hepatitis B or C virus (HBV, HCV), human immunodeficiency virus (HIV), or acquired immunodeficiency syndrome (AIDS).Poorly controlled metabolic disorders.Poorly controlled endocrine disorders.

In the group of patients with cardiovascular risk or cardiovascular disease, the following were included:Patients smoking for more than 5 years.Diabetes or impaired glucose tolerance.Family history of cardiovascular disease.Lack of physical activity or sedentary lifestyle.Unhealthy diet, high in saturated and trans fats, salt, and sugar.Sleep apnea or other sleep disorders.Obesity (body mass index over 30 [8]).Pre-existing hypertension in pregnancy (hypertension present before conception or detected at the latest 20 weeks of gestation [9]). Hypertensin definition: systolic blood pressure is higher than 140 mmHg and diastolic blood pressure is higher than 90 mmHg [10].Pregnancy-induced hypertension (hypertension known to be present after 20 weeks of gestation [11].Preeclampsia (hypertension plus proteinuria [12]).Eclampsia (seizures/coma with preeclampsia [13]).Low-density lipoprotein cholesterol > 190 mg/dL.Total cholesterol > 280 mg/dL.Triglycerides > 200 mg/dL.

Blood was collected through venipuncture for telomere length analysis in 2 mL vacutainer tubes containing EDTA as an anticoagulant. Blood tubes were immediately centrifuged, and white blood cells were separated and collected for DNA extraction, which was carried out using the Maxwell^®^ RSC Buffy Coat DNA Kit (Promega Corp, Madison, WI, USA) in a Maxwell^®^ RSC 48 Instrument (Promega Corp, Madison, WI, USA) according to the manufacturer’s instructions. The telomere length was assessed through real-time PCR using the Absolute Human Telomere Length Quantification qPCR Assay Kit (ScienCell, Carlsbad, CA, USA) in an ABI 7900HT real-time PCR instrument (Thermo Fischer Scientific, Waltham, MA, USA), according to the manufacturer’s instructions. The kit uses specific primer sets that recognize human telomere sequences, using SYBR Green to visualize and monitor the amplification. A single-copy reference primer set was also included, that recognizes and amplifies a 100 bp-long region on human chromosome 17 and serves as a reference for data normalization. A reference genomic DNA sample with a known telomere length served as a reference for calculating the telomere length of target samples using the comparative ΔΔCq (Quantification Cycle Value) method. Total telomere length (TTL) was expressed in megabases (MB)/diploid cell.

The data obtained from the study were analyzed using GraphPad Prism (version 5, GraphPad Software, Boston, MA, USA). Prior to analysis, data were screened for outliers, missing values, and normality. Variables were transformed or normalized as necessary to meet the assumptions of the statistical tests.

Descriptive statistics were used to summarize the data, and group differences were analyzed using *t*-tests. The *t*-test is a statistical method used to determine whether there is a significant difference between the means of two groups. It is typically used when the sample size is small. The test compares the means of two groups and determines whether the difference between them is statistically significant or simply due to chance.

All statistical tests were two-tailed, and *p*-values less than 0.05 were considered statistically significant. Results are presented as mean ± standard deviation (SD). Mean, also known as the average, is a statistical measure of central tendency. It is calculated by adding all the values in a set of data and dividing the sum by the total number of values. Standard deviation is a statistical measure of the amount of variability or dispersion in a set of data. It represents the degree of spread or variation of the data points around the mean or average value. Graphs were generated using GraphPad Prism and are presented with appropriate labeling and formatting.

## 3. Results

The patients were pregnant women aged between 18 and 40 years old (the mean age being 27.88 years old).

The data presented in Table 1 include the mean and standard deviation values for the two groups, Groups 1 and 2. Group 1 had a mean value of 0.5728 and a standard deviation of 0.5003, while Group 2 had a mean value of 0.3537 and a standard deviation of 0.3502. Table 2 includes the values of TTL MB for both Groups 1 and 2.

Furthermore, a *t*-test was conducted to determine if there was a significant difference between the mean telomere lengths of the two groups. The results indicated that there was a statistically significant difference between the mean telomere lengths of Groups 1 and 2 (*p*-value = 0.0458), indicating that the mean telomere length in Group 1 was significantly shorter than that of Group 2. Based on the presented data, there appeared to be a strong association between cardiovascular risk and telomere shortening.

Figure 1 illustrates the distribution of TTL for the two groups, with the mean values along and standard deviations displayed as a horizontal line for each group. The circular symbol signifies the quantification of telomere length, measured in megabases, pertaining to group 1, whereas the square symbol denotes the quantification of telomere length, also measured in megabases, specifically relating to group 2. The x-axis represents the range of TTL values, while the y-axis displays the frequency or the number of observations within each TTL range. The graph’s shape and spread demonstrate how the TTL values are distributed across the two groups. The mean values for each group are displayed as a reference point to compare the central tendency of the distributions. The visual representation indicates that Group 1 had a higher TTL range compared to Group 2. Additionally, the mean TTL value for Group 1 was higher than that of Group 2. The difference in the mean values is noticeable, suggesting a significant variation between the two groups. The graph provides a clear comparison of the TTL values and central tendency for the two groups, indicating that Group 1 had a higher TTL range and mean value than Group 2.

The length of telomeres in relation to patient age is depicted in the following two graphs. Figure 2 represents the age-related distribution of the telomere length values in Group 1, where the x-axis represents age categories (the circular symbol), and the y-axis displays the TTL. The age was measured in years. The graph shows a decreasing trend in telomere length values with increasing age in Group 1, indicating that the telomere length was influenced by age in this group. However, this trend was not evident in Group 2, as shown in Figure 3, indicating that age was not a significant factor in determining the telomere length in this group.

Figure 3 represents the age-related distribution of the telomere length values in Group 2, where the x-axis represents age categories age (the circular symbol), and the y-axis displays the TTL. The age was measured in years. The graph shows no significant decrease in the telomere length values with increasing age in Group 2. These findings suggest that cardiovascular risk was a more critical factor than age in determining the telomere length values in Group 2.

Together, the results from Figure 2 and Figure 3 suggest that the telomere length was more influenced by cardiovascular risk factors than age. While age-related decreases in telomere length were evident in Group 1, the absence of this trend in Group 2 suggests that cardiovascular risk was a more significant determinant of the telomere length values. These findings have important implications for the evaluation of telomere length changes and cardiovascular risk in different age groups.

A total of 32% of women who were pregnant and belonged to Group 2 gave birth to infants with intrauterine growth restriction, indicating a correlation between the shortened length of telomeres in mothers and intrauterine growth restriction.

## 4. Discussion

When a pregnant woman is diagnosed with a cardiovascular risk, it can cause significant concern and anxiety [14]. This is because such risks can potentially impact the health of both the mother and the developing fetus. It is therefore important to undertake a careful evaluation of the patient’s health status and any associated risk factors. This can involve monitoring blood pressure, blood sugar levels, cholesterol levels, and family history. By identifying and addressing any potential risk factors early on, healthcare professionals can work to ensure that the patient has the healthiest possible pregnancy outcome. Ultimately, the goal of this approach is to help reduce the likelihood of complications and ensure a safe and successful pregnancy for both the mother and the child.

Several studies have investigated the relationship between the telomere length and pregnancy-related complications. One study [15] found that women with a shorter telomere length were more likely to develop preeclampsia, a pregnancy-related condition characterized by high blood pressure and protein in the urine. The same study demonstrated that maternal telomere length is not related to severe preeclampsia but is negatively associated with gestational age and is also affected by race. In our study, we did not have patients from different ethnicities, and therefore, we could not establish an association between the telomere length and race. However, we did observe a distribution of low telomere length values in patients in the second half of the selected age range (closer to 40 years old than to 18 years old).

Placentas from pregnancies with preeclampsia and intrauterine growth restriction exhibited shorter telomeres.

The formation of the mother’s telomere aggregates is more pronounced in preeclampsia cases compared to intrauterine growth restriction, indicating distinct placental stress-induced mechanisms in preeclampsia with or without intrauterine growth restriction. Therefore, intrauterine growth restriction (IUGR), a condition in which a fetus does not adequately grow in the womb, has been linked to telomere shortening in both the mother and the child [16]. In our study, we demonstrated telomere shortening in patients with both preeclampsia and intrauterine growth restriction, but we did not measure the child’s telomere length.

Another study reported that women with a history of preeclampsia had a shorter telomere length than women who had uncomplicated pregnancies [15]. These findings and our findings suggest that a shortened telomere length may play a role in the development of preeclampsia and may serve as a biomarker for identifying women at an increased risk of developing this condition.

Pregnancy-induced hypertension has been found to cause accelerated aging of the cardiovascular system and a wider range of cardiovascular conditions, including valvular heart disease, than previously thought. The risk of cardiovascular disease after pregnancy-induced hypertension is primarily influenced by the development of chronic hypertension, although this relationship is not fully understood [17].

A study has shown convincing evidence of the association between a shortened telomere length and a higher risk of coronary atherosclerosis, myocardial infarction, ischemic heart disease, and stroke, as well as a lower risk of hypertension. However, they did not observe any causal links between telomere length and heart failure, atrial fibrillation, and cardiac death. Further research is needed to verify these findings and explore the mechanisms that underlie these relationships [4].

Numerous studies have examined the association between telomere length and complications related to pregnancy [18]. Nonetheless, the precise mechanisms involved in this relationship are not yet fully elucidated. One postulated hypothesis is that telomere shortening may lead to pregnancy-related complications by inducing endothelial dysfunction and inflammation [19]. Telomere malfunctioning has been associated with oxidative stress, which can cause damage to the endothelial cells lining blood vessels and impede their capacity to produce nitric oxide, a crucial mediator of vascular function [20].

Telomere length has potential as a biomarker for identifying women at increased risk of pregnancy-related complications and may have therapeutic implications. Telomerase, the enzyme responsible for maintaining the telomere length, has been the target of several experimental therapies for age-related diseases. While these therapies are in the early stages of development, they may have the potential for preventing or treating pregnancy-related complications [21].

Maternal nutrition plays a crucial role in fetal growth and development, as well as pregnancy outcomes [22]. Women with preeclampsia exhibit changes in maternal nutrition and heightened oxidative stress levels. As oxidative stress has been linked to a shortened telomere length and the formation of shortened telomere aggregates, increased telomere attrition may result in heightened cellular senescence and tissue aging.

Recent research has shown that certain medications can affect the length of telomeres. In this regard, in this study, we excluded the patients who have taken these medications throughout their lifetime. Chemotherapy drugs are used to treat cancer by killing rapidly dividing cancer cells. However, these drugs can also affect healthy cells, including those with telomeres. Some chemotherapy drugs, such as cyclophosphamide and doxorubicin, have been found to cause telomere shortening. The exact mechanisms behind chemotherapy-induced telomere shortening are not yet fully understood, but it is thought to be due in part to the generation of reactive oxygen species, which can damage DNA and contribute to telomere shortening. Additionally, chemotherapy drugs can affect the activity of telomerase, an enzyme that helps to maintain the telomere length. When telomerase activity is disrupted, telomeres can shorten more quickly. Overall, the effects of chemotherapy drugs on telomere length are an area of active research, as understanding these effects may help to improve cancer treatment and reduce the risk of long-term complications associated with chemotherapy.

Some chronic illnesses can impact the telomere length. Therefore, one of the inclusion criteria was that patients should not have any of the conditions that have the potential to shorten the telomere length. Telomere shortening has been observed in many types of cancer cells, including lung cancer [23], breast cancer [24], and colorectal cancer [25]. The relationship between telomere shortening and cancer is complex and is an area of active research. While telomere shortening is a common feature of many types of cancer cells, it is not clear whether telomere shortening is a cause or a consequence of cancer. Additionally, some studies have suggested that telomerase inhibitors may have potential as anticancer therapies [26], as they may be able to specifically target cancer cells and prevent them from rebuilding their telomeres. However, further research is needed to fully understand the role of telomeres in cancer and to develop effective treatments that target telomere-related pathways.

Several studies have shown that individuals with type 2 diabetes tend to have shorter telomeres compared to healthy individuals [27]. One possible explanation for this is that high blood sugar levels may cause oxidative stress, which can damage telomeres and shorten their length.

Studies have found that in individuals with shorter telomeres, there may be an associated with cognitive impairment and dementia [28], including Alzheimer’s disease [29]. One potential mechanism for this link is oxidative stress. Oxidative stress can damage telomeres and lead to accelerated telomere shortening, which in turn may contribute to cognitive decline and neurodegeneration. Additionally, chronic inflammation and immune system dysfunction, which have also been linked to telomere shortening, may contribute to the development of neurodegenerative disorders. Furthermore, research has shown that the telomere length may be affected by lifestyle factors such as exercise, diet, and stress management, which may also impact the risk of neurodegenerative disorders. For instance, regular exercise has been associated with longer telomeres and a reduced risk of cognitive decline, while chronic stress has been linked to telomere shortening and an increased risk of Alzheimer’s disease.

Some studies have suggested that the telomere length is linked to depression [30], with individuals who experience depression having shorter telomeres compared to those who do not. One potential mechanism for this link is chronic stress. Depression is often associated with chronic stress, which has been shown to contribute to telomere shortening. In addition, chronic inflammation, which is also linked to telomere shortening, may contribute to the development of depression.

Furthermore, studies have consistently shown that a short telomere length is associated with a higher mortality risk [31]. In fact, the telomere length has been identified as a potential biomarker for aging and age-related diseases, including cardiovascular disease, cancer, and neurodegenerative disorders. A short telomere length may contribute to mortality risk through several mechanisms. First, telomere shortening is associated with cellular aging and a decreased regenerative capacity, which may lead to the accumulation of damage and dysfunction in tissues and organs over time. This can contribute to the development of age-related diseases, and ultimately, mortality.

A shorter telomere length has also been found to be associated with decreased immune system function and an increased risk of infections [32]. It is thought that the telomere length may be an important indicator of overall cellular health and a potential biomarker for various age-related diseases. However, more research is needed to fully understand the relationship between the telomere length and these diseases.

Strengths and Limitations:

One of the major strengths of this study is its focus on a specific population and outcome. The study targeted high-risk pregnant women, which allowed the researchers to examine a unique population that is at increased risk for cardiovascular disease and other adverse health outcomes. The study also focused on a specific outcome—telomere shortening—which is an important biomarker of aging and a potential predictor of future disease risk.

The prospective design of the study is another strength. By following the participants over time, the researchers were able to establish cause-and-effect relationships and observe changes in the outcome of interest.

The use of validated measures for both the exposure and outcome is also a strength. The researchers used well-established measures to assess cardiovascular risk and the telomere length, which enhanced the reliability and validity of the results. The study also employed appropriate statistical methods to analyze the data, ensuring that the results were robust and reliable.

The study’s novelty is another strength. This is a relatively unexplored area of research, and the study contributes to the growing body of literature in this field. The study’s findings have the potential to inform future research and clinical practice in this area.

Overall, the study’s strengths are significant and demonstrate the potential for future research in this area. The focus on a specific population and outcome, the prospective design, the use of validated measures, appropriate statistical analysis, and novelty all contribute to the value of this study. Despite its limitations, the study’s strengths provide important insights into the potential relationship between cardiovascular risk and telomere shortening in pregnant women.

One of the major limitations of the study is the small sample size. The study was conducted on a limited number of patients, in a single medical center, over a period of three years. This small sample size may limit the generalizability of the findings to a broader population. A larger study conducted over a longer period of time with a more diverse sample may provide more conclusive results.

Another limitation is the potential for selection bias. The participants were recruited from a single medical center and may not be representative of the broader population. Furthermore, the study excluded patients with certain pre-existing conditions, which may have influenced the results.

There is also a potential for confounding variables to influence the results. For example, the study did not account for other factors that may influence the telomere length, such as stress, diet, and lifestyle factors. Additionally, the study did not account for the possibility of reverse causation, where short telomeres may be a consequence of poor health rather than a cause.

Another limitation is the potential for measurement errors. While the study used validated measures for both the exposure and the outcome, there may still be errors in the measurement of these variables. For example, the measurement of the telomere length may be influenced by factors such as the timing of the blood draw or the method used to measure the telomere length.

In conclusion, while our study has several strengths, it also has several limitations that need to be considered when interpreting the results. The small sample size, potential for selection bias, confounding variables, and measurement errors all limit the generalizability and causal interpretation of the findings. Further research is needed to confirm and extend the results of this study.

## 5. Conclusions

The findings of this study suggested that pregnant women with cardiovascular risk have a shorter telomere length compared to those without cardiovascular risk. The telomere length could be a potential marker of cardiovascular risk during pregnancy, and this finding supports the need for increased surveillance and management of cardiovascular risk during pregnancy.

In conclusion, the telomere length has emerged as a promising biomarker for predicting the pregnancy-related CVD risk. Several studies have demonstrated an association between a shortened telomere length and an increased risk of pregnancy-related complications, such as preeclampsia, gestational diabetes, and IUGR. Further research is needed to elucidate the mechanisms underlying this relationship and to determine the clinical utility of the telomere length as a biomarker for predicting the pregnancy-related CVD risk. Nevertheless, the potential therapeutic implications of telomere-targeted therapies make this an exciting area of research for both maternal and fetal health.

## Figures and Tables

**Figure 1 medicina-59-01012-f001:**
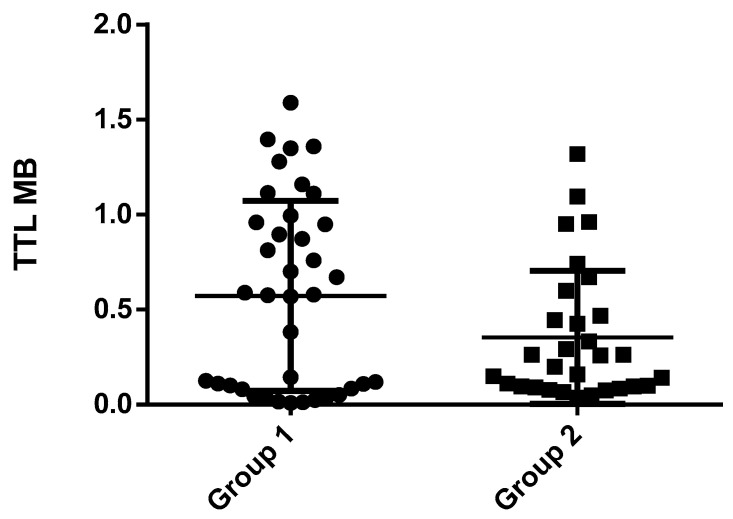
Distribution of TTL with mean values for the two groups. The circular symbol signifies the quantification of telomere length, measured in megabases, pertaining to group 1, whereas the square symbol denotes the quantification of telomere length, also measured in megabases, specifically relating to group 2.

**Figure 2 medicina-59-01012-f002:**
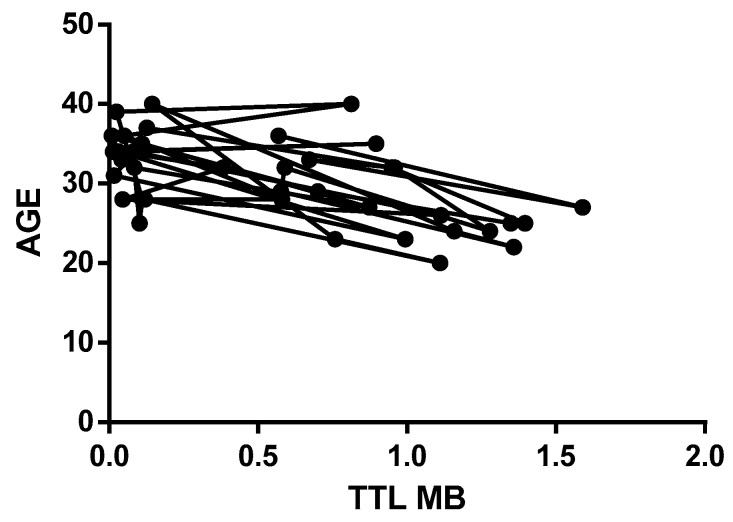
Distribution of the telomere length values in Group 1. Circular symbol: age categories.

**Figure 3 medicina-59-01012-f003:**
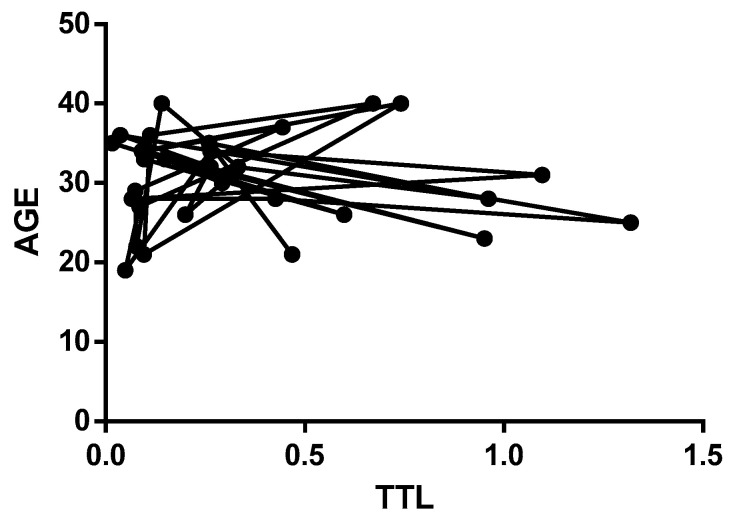
Distribution of the telomere length values in Group 2. Circular symbol: age categories.

**Table 1 medicina-59-01012-t001:** Statistical comparison of mean and standard deviation (SD) using the *t*-test.

	Group 1	Group 2
Mean	0.5728	0.3537
SD	0.5003	0.3502
*p*-value, *t*-test	0.0458	

**Table 2 medicina-59-01012-t002:** Values of Total telomere length (TTL) in megabases (MB) in Group 1 and Group 2.

Patients from Group 1	TTL MB	Patients from Group 2	TTL MB
1	0.1264268	1	0.467853
2	0.948867	2	0.259355
3	1.39604	3	0.961155
4	1.348471	4	0.332492
5	0.700402	5	0.14051
6	0.0086322	6	0.076892
7	0.041775	7	0.074098
8	0.0129339	8	0.443492
9	0.895752	9	0.098643
10	0.0806034	10	0.951142
11	1.114253	11	0.148464
12	0.0446673	12	0.291809
13	0.382372	13	0.199484
14	0.0309549	14	0.262581
15	0.993644	15	0.036069
16	0.0165901	16	1.31885
17	0.1090447	17	0.426514
18	0.872493	18	0.065456
19	0.0844323	19	1.096476
20	0.1012656	20	0.262663
21	0.1108394	21	0.048773
22	0.0236789	22	0.084439
23	0.813139	23	0.671653
24	0.0501761	24	0.111055
25	0.579271	25	0.095666
26	0.119228	26	0.740913
27	1.110441	27	0.090285
28	0.7584728	28	0.0958953
29	0.1435491	29	0.5986328
30	1.1586174	30	0.0158965
31	1.3587419		
32	0.5761285		
33	0.5894358		
34	1.2783495		
35	0.9583496		
36	0.6713856		
37	1.5896178		
38	0.5683498		

## Data Availability

The data presented in this study are available on request from the corresponding author.

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
