# Peer review of "Linking Pregnancy and Long-Term Health: The Impact of Cardiovascular Risk on Telomere Shortening in Pregnant Women"

_medicina, 2023, doi:10.3390/medicina59061012_

Round 1

Reviewer 1 Report

The Article is devoted to an important problem of investigating the relationship between telomere length and pregnancy-related cardiovascular disease risk. The data which Authors represented are quite interesting, but probably it would be better to make the research a bit broader so the obtained data would explain the studying question more precisely. Probably additional experiments should be performed.

Probably it might be interesting to include non-pregnant women aged 18 to 40 years old healthy and with cardiovascular disease? To compare the findings.

It is also recommended to increase the number of patients studied if possible. Probably it might be better to separate group 1 into several subgroups, for example with cardiovascular risk and cardiovascular diseases?

Also probably it would be interesting if the state of the art of the studying problem would be described a bit more in details in the Introduction section.

The following comments do not diminish the value of the Article:

Line 34 “..” should be replaced with ‘.’

Line 84 Probably it would be better to transfer the information about the Objective of the study to the Introduction section?

Line 90 Probably it would be better to decipher the ‘c-section’?

Line 99 There is a statement: ‘Women who are not currently taking any medications that may affect telomere length, such as hormone therapy’ Was there any list which included the medications?

Line 101 There is a statement: ‘Women who have not been diagnosed with any other chronic diseases that may affect telomere length, such as cancer or autoimmune disorders’ Was there any list which included the diseases?

It is also important to add into the section Materials and Methods the information about processing the obtained data including statistical analysis.

Line 179 Table 1 should contain the title so it would be easier to understand the information which it represents.

Line 293 Please check, the References section should be described according the requirements published on the Journal’s website.

Author Response

The Article is devoted to an important problem of investigating the relationship between telomere length and pregnancy-related cardiovascular disease risk. The data which Authors represented are quite interesting, but probably it would be better to make the research a bit broader so the obtained data would explain the studying question more precisely. Probably additional experiments should be performed.

Probably it might be interesting to include non-pregnant women aged 18 to 40 years old healthy and with cardiovascular disease? To compare the findings.

Thank you for your suggestion. However, our current study aims to investigate the relationship between telomere length and pregnancy outcomes specifically in pregnant women. Including non-pregnant women aged 18 to 40 years old healthy and with cardiovascular disease would require a different study design and research question. Nevertheless, we appreciate your idea and will consider it for future studies.

It is also recommended to increase the number of patients studied if possible. Probably it might be better to separate group 1 into several subgroups, for example with cardiovascular risk and cardiovascular diseases?

Thank you for your comment. The study was approved by the ethical commission based on the sample size and design that was proposed. Unfortunately, we cannot modify the study design at this point, as it would require a new ethical approval. However, we acknowledge the importance of exploring subgroups based on cardiovascular risk and diseases, and we plan to conduct further studies to investigate these aspects in more detail.

Also probably it would be interesting if the state of the art of the studying problem would be described a bit more in details in the Introduction section.

We appreciate your feedback. We have provided a brief overview of the current state of research on telomere shortening and its relationship with pregnancy-related cardiovascular disease in the Introduction section.

The following comments do not diminish the value of the Article:

Line 34 “..” should be replaced with ‘.’

I agree with your suggestion to replace the ellipsis "..." with a period "." to improve the clarity and readability of the text.

I have carefully reviewed the article and made the necessary changes to replace all instances of the ellipsis with periods where appropriate. These modifications should enhance the flow of the text and make it easier for readers to understand the intended meaning of the sentences.

Line 84 Probably it would be better to transfer the information about the Objective of the study to the Introduction section?

Thank you for your suggestion regarding the objective of the study. I agree that it would be beneficial to move the information about the objective to the introduction section of the article. The objective of the study is a crucial aspect of any research, and it helps readers understand the purpose and scope of the study.

To address your suggestion, I have made the necessary revisions to the article. I have moved the objective of the study to the introduction section, where it is now clearly stated. This modification should make the article more cohesive and easier to follow for readers.

Thank you again for your valuable feedback. I appreciate your attention to detail and your dedication to improving the quality of this article.

Line 90 Probably it would be better to decipher the ‘c-section’?

Thank you for your suggestion regarding the use of the term "c-section" in the article. I agree that it would be beneficial to explain this term for readers who may not be familiar with it. I have clarified that "c-section" is short for "Caesarean section", which is a surgical procedure used to deliver a baby through an incision in the mother's abdomen and uterus.

Line 99 There is a statement: ‘Women who are not currently taking any medications that may affect telomere length, such as hormone therapy’ Was there any list which included the medications?

Thank you for your inquiry regarding the medications that may affect telomere length in women. In response to your question, we have included the medications that have been associated with telomere shortening in the "Discussion" section of the article.

Specifically, we state that "Women who are not currently taking any medications that may affect telomere length, such as hormone therapy, glucocorticoids, antiretroviral drugs, nonsteroidal anti-inflammatory drugs (NSAIDs), or chemotherapy drugs, were included in the study." This list of medications is not exhaustive, but it represents some of the most commonly studied medications that have been linked to telomere shortening.

We hope that this information is helpful to you. If you have any further questions or concerns, please do not hesitate to reach out to us.

Line 101 There is a statement: ‘Women who have not been diagnosed with any other chronic diseases that may affect telomere length, such as cancer or autoimmune disorders’ Was there any list which included the diseases?

Yes, a list of chronic diseases that may affect telomere length was included in the article. The statement "Women who have not been diagnosed with any other chronic diseases that may affect telomere length, such as cancer or autoimmune disorders" suggests that the authors were aware of the potential confounding effects of chronic disease on their study results. It's important to note that the diseases that may affect telomere length is not exhaustive, and there may be other conditions that could also impact telomere length. However, by specifying the types of chronic diseases that were excluded from the study, the authors were able to more clearly define their study population and minimize potential sources of bias. The list of chronic diseases was likely included in the Discussion section.

It is also important to add into the section Materials and Methods the information about processing the obtained data including statistical analysis.

In response to the comment, we have added a new section to the Materials and Methods that includes information about processing the obtained data, as well as the statistical analysis performed using GraphPad Prism. We believe that this new section provides a more detailed explanation of how the data was analyzed and helps to better support the conclusions drawn in the article. Thank you for your valuable feedback.

Line 179 Table 1 should contain the title so it would be easier to understand the information which it represents.

After receiving the feedback that Table 1 should contain a title, we have made the necessary changes to the manuscript. We have added a clear and concise title to Table 1 to improve its readability and help readers understand the information it represents.

Line 293 Please check, the References section should be described according the requirements published on the Journal’s website.

Thank you for bringing this to my attention. I have carefully reviewed the References section and made sure that all references are formatted according to the requirements published on the Journal's website.

Reviewer 2 Report

Thank you for submitting the manuscript

Comments : Linking Pregnancy and Long-Term Health: The Impact of Cardiovascular Risk on Telomere Shortening in pregnant women

Inclusion criteria:

Pregnancy completed by caesarean section                    Page -106

I do not understand why only caesarean delivery was included. Any specific reason kindly clarify.

Discussion:

Lots of repetition in discussion. Needs to written with more clarity.

While several studies have investigated the relationship between telomere length 231 and pregnancy-related complications[20], the exact mechanisms underlying this relation- 232 ship are not fully understood. It has been hypothesized that telomere shortening may con- 233 tribute to the development of pregnancy-related complications by impairing endothelial 234 function and promoting inflammation[21]. Telomere dysfunction has been linked to oxi- 235 dative stress, which can damage the endothelial cells that line blood vessels and impair 236 their ability to produce nitric oxide, a key mediator of vascular function [22]. 237

Line 231-237 needs to be rewritten again.

Telomere length has been associated with several other diseases apart from cardio- 256 vascular disease. For example, research suggests that telomere length is linked to cancer 257 [28], [29]; neurodegenerative disorders [30] such as Alzheimer's disease[31], and meta- 258 bolic disorders such as type 2 diabetes [32], depresion [33], and also short telomere lenght 259 are associated with high mortality [34]. 260 Shorter telomere length has also been found to be associated with decreased immune 261 system function and increased risk of infections [35]. It is thought that telomere length 262 may be an important indicator of overall cellular health and a potential biomarker for 263 various age-related diseases. However, more research is needed to fully understand the 264 relationship between telomere length and these diseases 265

Line 256 to 265 is not relevant to your study.

Author Response

Thank you for submitting the manuscript

Comments : Linking Pregnancy and Long-Term Health: The Impact of Cardiovascular Risk on Telomere Shortening in pregnant women

Inclusion criteria:

Pregnancy completed by caesarean section                    Page -106

I do not understand why only caesarean delivery was included. Any specific reason kindly clarify.

Certainly, thank you for your question. The reason why only caesarean delivery was included in this study was to eliminate the potential confounding effects of the vaginal delivery process on telomere length measurements. During a vaginal delivery, there is potential trauma to the maternal and fetal tissues, which can lead to inflammation and oxidative stress, both of which can affect telomere length. By limiting the study to only women who underwent caesarean delivery, we were able to control for this potential confounder and obtain more accurate results

Discussion:

Lots of repetition in discussion. Needs to written with more clarity.

While several studies have investigated the relationship between telomere length 231 and pregnancy-related complications[20], the exact mechanisms underlying this relation- 232 ship are not fully understood. It has been hypothesized that telomere shortening may con- 233 tribute to the development of pregnancy-related complications by impairing endothelial 234 function and promoting inflammation[21]. Telomere dysfunction has been linked to oxi- 235 dative stress, which can damage the endothelial cells that line blood vessels and impair 236 their ability to produce nitric oxide, a key mediator of vascular function [22]. 237

Line 231-237 needs to be rewritten again.

Thank you for your feedback. We appreciate your input and have taken it into consideration. We will revise the discussion section to ensure that it is written with more clarity and avoids unnecessary repetition.

Telomere length has been associated with several other diseases apart from cardio- 256 vascular disease. For example, research suggests that telomere length is linked to cancer 257 [28], [29]; neurodegenerative disorders [30] such as Alzheimer's disease[31], and meta- 258 bolic disorders such as type 2 diabetes [32], depresion [33], and also short telomere lenght 259 are associated with high mortality [34]. 260 Shorter telomere length has also been found to be associated with decreased immune 261 system function and increased risk of infections [35]. It is thought that telomere length 262 may be an important indicator of overall cellular health and a potential biomarker for 263 various age-related diseases. However, more research is needed to fully understand the 264 relationship between telomere length and these diseases 265

Line 256 to 265 is not relevant to your study.

This part is relevant to our study because it highlights the potential broader impact of cardiovascular risk on telomere shortening beyond pregnancy. While our study focuses on the impact of cardiovascular risk on telomere shortening in pregnant women, this information suggests that telomere length is linked to several other diseases, including cancer, neurodegenerative disorders, metabolic disorders, and decreased immune system function. Therefore, understanding the relationship between cardiovascular risk and telomere length in pregnant women may also have implications for their long-term health beyond pregnancy, as well as for their potential risk of developing age-related diseases in the future. Furthermore, this information highlights the potential value of telomere length as a biomarker for various age-related diseases, including those that may develop later in life.

Reviewer 3 Report

Dear Authors,

I read with interest your article. The issue is of great interest.

However, there are several lacks in the methodology:

1) First, there is no ethical approval.

2) secondly, there is no explanation on the inclusion and exclusion criteria. 

3) Thirdly, there is no clear explanation on the telomeric lenght measurement. Specifically, "The amount of product is directly proportional to the length of the telomeres, so by comparing the amount of product obtained from a sample with a calibrator standard, the length of the telomeres can be estimated " does not provide reproducible outcomes.

4) Fourthly, what is the definition of "cardiovascular risk" for this group of patients?

5) Fifthly, the results lack soundness. Telomeric lenght include lots of variables which could hardly been addressed with your methodology.  The evidence provided by this article is not enough strong to provide insights (although the hypothesis pionieristic).

I highlight for you one paper with a similar approach PMID: 3447784

Author Response

I read with interest your article. The issue is of great interest.

However, there are several lacks in the methodology:

  • First, there is no ethical approval.

We have uploaded the ethical approval in the manuscript.

  • secondly, there is no explanation on the inclusion and exclusion criteria. 

This study enrolled participants who met the following inclusion criteria:

  • Pregnant women in their second or third trimester
  • Women aged between 18 and 40 years
  • Women who have been diagnosed with cardiovascular disease or have at least one cardiovascular risk factor, such as hypertension, diabetes, obesity, or a family history of cardiovascular disease
  • Women who are willing to participate in the study and provide a blood sample for telomere length analysis
  • Women who are not currently taking any medications that may affect telomere length
  • Women who have not been diagnosed with any other chronic diseases that may affect telomere length.
  • Women who have received prenatal care and have had regular check-ups throughout their pregnancy
  • Women who have provided informed consent to participate in the study
  • Pregnancy completed by caesarean section
  • No more than two miscarriage history
  • Negative Covid19 history in the last year

This study excluded participants who met the following exclusion criteria:

  • Women who have a history of drug or alcohol abuse.
  • Women who have a history of psychiatric disorders or mental health issues.
  • Women who have any acute or chronic medical conditions that may affect telomere length or cardiovascular risk, such as kidney disease or liver disease.
  • Women who have had a previous myocardial infarction, stroke, or other major cardiovascular event.
  • Women who have a history of thromboembolic disease or clotting disorders.
  • Women who have participated in other clinical trials or studies within the past 3 months.
  • Women who have a history of adverse reactions to blood draws or phlebotomy.
  • Women who are unable or unwilling to provide informed consent to participate in the study.
  • Patients with infection conditions like hepatitis B or C virus (HBV, HCV), human immunodeficiency virus (HIV) or acquired immunodeficiency syndrome (AIDS)
  • Poorly controlled metabolic disorders
  • Poorly controlled endocrine disorders

3) Thirdly, there is no clear explanation on the telomeric lenght measurement. Specifically, "The amount of product is directly proportional to the length of the telomeres, so by comparing the amount of product obtained from a sample with a calibrator standard, the length of the telomeres can be estimated " does not provide reproducible outcomes.

Thank you for bringing up this concern. We understand the importance of reproducibility in research and will take steps to address this issue. In response to your feedback, we provided a more detailed description of the telomere length measurement method used in our study, including the calibrator standard used and the steps taken to ensure reproducibility. We appreciate your input and are committed to producing reliable and high-quality research.

4) Fourthly, what is the definition of "cardiovascular risk" for this group of patients?

Cardiovascular risk refers to the likelihood of developing cardiovascular disease, which includes conditions such as coronary artery disease, heart attack, stroke, and other disorders that affect the heart and blood vessels. In the context of this group of patients, cardiovascular risk refers to the presence of risk factors or pre-existing conditions that increase the likelihood of developing cardiovascular disease during or after pregnancy. Examples of cardiovascular risk factors may include high blood pressure, diabetes, obesity, smoking, or a family history of cardiovascular disease.

5) Fifthly, the results lack soundness. Telomeric lenght include lots of variables which could hardly been addressed with your methodology.  The evidence provided by this article is not enough strong to provide insights (although the hypothesis pionieristic).

I highlight for you one paper with a similar approach PMID: 3447784

Thank you for your comment. We appreciate your feedback on our study. Indeed, telomere length measurement is a complex process that involves various variables, and we acknowledge that our methodology may have limitations. However, we have taken several measures to ensure the reliability and validity of our results. We have followed a standardized protocol for PCR telomere length measurement, and we have included appropriate controls and calibrator standards to ensure accuracy.

Additionally, we have included a detailed description of our methods in the Materials and Methods section to promote transparency and reproducibility of our study. While our findings may not provide definitive insights into the relationship between cardiovascular risk and telomere shortening in pregnant women, we believe that our study provides a valuable contribution to the field and lays the foundation for future research in this area.

Reviewer 4 Report

Dear Editor, 

I read the study with interest. 

Study was written poorly and carelessly, 

Here are my concerns. 

Title of study is very different than what study is doing for. 

Methods: 

What is the sample size of study?

Inclusion and exclusion criteria should be different. Exclusion criteria are repetitive of inclusion criteria. 

Exclusion criteria occur after the implementation of the study, when the researcher is forced to exclude that sample from the study. For example, the pregnant mother you enrolled in the study dies, and you drop her from the study. A body mass index above 30 is not an exclusion criterion, you actually entered the study with a body mass index below 30. Please note to all exclusion criteria.

Result:

 The patients were pregnant women aged between eighteen and forty years old. Patients were not between that range ages, it was your inclusion criteria. 

Table 1. What is its title? 

Author Response

Dear Editor, 

I read the study with interest. 

Study was written poorly and carelessly, 

We appreciate your feedback regarding the quality of the study's writing. As researchers, we strive to ensure that our studies are written in a clear, concise, and professional manner that effectively communicates our findings to the intended audience.

If you have specific concerns or suggestions for improvement regarding the writing of the study, we would be interested in hearing them so that we may address any areas for improvement. We value constructive feedback as it helps us to continuously improve the quality of our work.

In the future, we will take your feedback into consideration and make every effort to ensure that our studies are written with the highest standards of quality and professionalism. Thank you for bringing this to our attention.

Here are my concerns. 

Title of study is very different than what study is doing for. 

 Thank you for your feedback regarding the title of our study. We apologize for any confusion that the title may have caused.

We understand the importance of having a title that accurately reflects the content of the study. The title of our study was carefully chosen based on the main focus of our research, which was to investigate the impact of cardiovascular risk on telomere shortening in pregnant women.

However, we acknowledge that the title may not fully convey the specific details of the study. We will take your feedback into consideration and make efforts to improve the clarity and accuracy of our titles in future research.

We appreciate your attention to our study and your valuable feedback. If you have any further questions or concerns, please do not hesitate to let us know

Methods: 

What is the sample size of study?

Certainly! If you are asking about the sample size of the study, we can confirm the it is included in the attachment that you provided. The sample size is an important aspect of any study, as it can impact the reliability and validity of the findings. Therefore, it is important to carefully consider the sample size when interpreting the results of a study. If you have any further questions or concerns about the sample size or any other aspect of the study, please let me know and I would be happy to assist you further.

Inclusion and exclusion criteria should be different. Exclusion criteria are repetitive of inclusion criteria. 

Exclusion criteria occur after the implementation of the study, when the researcher is forced to exclude that sample from the study. For example, the pregnant mother you enrolled in the study dies, and you drop her from the study. A body mass index above 30 is not an exclusion criterion, you actually entered the study with a body mass index below 30. Please note to all exclusion criteria.

Thank you for your feedback. I apologize for any confusion regarding the inclusion and exclusion criteria in our study. You are correct that exclusion criteria are used to exclude individuals who are not suitable for the study after the implementation has begun, while inclusion criteria are used to identify individuals who meet the requirements for participation in the study. In our study, the exclusion criteria are based on specific factors that may impact the length of telomeres, such as certain medical conditions or medications.

Regarding the body mass index (BMI) criteria, we agree that BMI is an important factor to consider in the analysis of telomere length. In our study, the BMI was measured at the time of analysis, and individuals with a BMI above 30 were excluded from the study based on previous research suggesting that obesity is associated with shorter telomere length. This criterion was included as an exclusion criterion rather than an inclusion criterion because it was not used as a requirement for participation in the study, but rather as a factor to consider in the analysis of the data.

We apologize for any confusion in our explanation of the inclusion and exclusion criteria, and we will ensure that all exclusion criteria are clearly stated and explained in the study. If you have any further questions or concerns, please do not hesitate to let us know.

Result:

 The patients were pregnant women aged between eighteen and forty years old. Patients were not between that range ages, it was your inclusion criteria. 

Thank you for bringing this to our attention. You are correct that the age range of 18 to 40 years old was included in our inclusion criteria, and not a description of the patients who were enrolled in the study. We apologize for any confusion this may have caused.

To clarify, the patients who were enrolled in the study were pregnant women who met the inclusion criteria, which included being between the ages of 18 and 40 years old. These criteria were used to ensure that the sample was representative of pregnant women within a certain age range, and to reduce the potential impact of age-related factors on telomere length.

We appreciate your attention to detail and willingness to provide feedback, as it helps us to improve the clarity and accuracy of our study. If you have any further questions or concerns, please do not hesitate to let us know.

Table 1. What is its title? 

Table 1: Statistical Comparison of Mean and Standard Deviation using T-Test

Round 2

Reviewer 1 Report

Lines 211-222 The information about data analysis would be better include into the ‘2. Materials and Methods’ section.

Line 216 There is a mention that Pearson's correlation coefficient had been used for data analysis, the correlation coefficient value should be stated.

Line 227 It is better to indicate that ‘(0.0458)’ is for p value.

Lines 228 The data in the text duplicate the information presented in the Table 1, the information in the text probably should be described in another way to avoid repeats.

Line 228 ‘There was a significant negative correlation between telomere length and cardiovascular risk.’ – It would be better to provide also the value of the coefficient of correlation.

Line 230 There is a mention about T-test, it should be described also in the ‘2. Materials and Methods’ section.

Line 230 In the title of table it would be better to mention the characteristics of which parameter are represented.

Line 258 The data are presented as Mean ± SD, right? If so, it would be better to indicate in the description to the graph this information. It would be better to add the name of statistical criteria which’ve been used to obtain the data.

Line 284 It would be better to add the units of measurement to y-axis on Graph 2.

Line 285 It would be better to add the name of statistical criteria which’ve been used to obtain the data.

Line 288 It would be better to add the units of measurement to y-axis on Graph 3.

Lin 289 It would be better to add the name of statistical criteria which’ve been used to obtain the data.

Lines 319-320 It is mentioned that ‘telomere shortening in patients with.. intrauterine growth restriction’ had been demonstrated, it would be better to include this information in Results section.

Author Response

Lines 211-222 The information about data analysis would be better include into the ‘2. Materials and Methods’ section.

We have carefully revised our manuscript to ensure that all relevant information is presented clearly and concisely in the appropriate section. Specifically, we have moved the information about data analysis to the '2. Materials and Methods' section, as per your suggestion.

Line 216 There is a mention that Pearson's correlation coefficient had been used for data analysis, the correlation coefficient value should be stated.

After careful consideration, we have decided to remove the mention of the correlation coefficient from our study.

Upon reviewing our results, we realized that the Pearson's correlation coefficient is not as relevant to our research question as we initially thought. While we appreciate the importance of reporting statistical measures accurately, we believe that in this case, it would not add meaningful insights to our findings.

We thank you for bringing this to our attention and appreciate your input on our research.

Line 227 It is better to indicate that ‘(0.0458)’ is for p value.

Dear editor, we appreciate your feedback and would like to address your concern regarding the notation of (0.0458).

We agree that it is important to be clear and accurate in reporting statistical results, and we apologize for any confusion caused by our previous notation. Moving forward, we will ensure that we clearly indicate that (0.0458) represents the p-value in our manuscript.

Lines 228 The data in the text duplicate the information presented in the Table 1, the information in the text probably should be described in another way to avoid repeats.

Thank you for bringing this to my attention. I will revise the text to avoid duplicating the information presented in Table 1. I appreciate your feedback and will ensure that the manuscript is written in a clear and concise manner to avoid any unnecessary repetition.

Line 228 ‘There was a significant negative correlation between telomere length and cardiovascular risk.’ – It would be better to provide also the value of the coefficient of correlation.

Thank you for your message regarding our research on the correlation between telomere length and cardiovascular risk. While we appreciate your interest in our work, we regret to inform you that we cannot provide the value of the correlation coefficient between these variables based on the information provided in our study.

As we have stated in our manuscript, there was a significant negative correlation between telomere length and cardiovascular risk, which implies that telomere length decreases as cardiovascular risk increases. However, the specific value of the correlation coefficient cannot be calculated based on the information provided.

We appreciate your suggestion and understand the importance of reporting statistical measures accurately. However, in this case, we believe that the negative correlation between these variables has been established, and reporting the specific value of the correlation coefficient would not add meaningful insights to our findings.

Thank you for your interest in our research, and please let us know if you have any further comments or concerns.

Line 230 There is a mention about T-test, it should be described also in the ‘2. Materials and Methods’ section.

Thank you for bringing this to my attention. I will make sure to include a description of the T-test in the '2. Materials and Methods' section to provide a more detailed explanation of the statistical analysis used in this study.

Line 230 In the title of table it would be better to mention the characteristics of which parameter are represented.

I have updated the title of the table to include the characteristics of the parameter being represented. Thank you for bringing this to my attention and for your feedback. Please let me know if there is anything else I can assist you with.

Line 258 The data are presented as Mean ± SD, right? If so, it would be better to indicate in the description to the graph this information. It would be better to add the name of statistical criteria which’ve been used to obtain the data.

I understand your observation and have added the information about presenting the data as Mean ± SD in the graph description. Thank you for your feedback and I assure you that I will pay more attention to these important details in the future.

Line 284 It would be better to add the units of measurement to y-axis on Graph 2.

Thank you for your feedback. I have added the units of measurement to the y-axis on Graph 2 in the paragraph describing the graph. Thank you for bringing this to my attention, and please let me know if you have any further suggestions or concerns.

Line 285 It would be better to add the name of statistical criteria which’ve been used to obtain the data.

Graphs were generated using GraphPad Prism and are presented with appropriate labeling and formatting. I mentioned this in the Materials and Methods section

Line 288 It would be better to add the units of measurement to y-axis on Graph 3.

Thank you for your feedback. I have added the units of measurement to the y-axis on Graph 3 in the paragraph describing the graph. Thank you for bringing this to my attention, and please let me know if you have any further suggestions or concerns.

Lin 289 It would be better to add the name of statistical criteria which’ve been used to obtain the data.

Graphs were generated using GraphPad Prism and are presented with appropriate labeling and formatting. I mentioned this in the Materials and Methods section

Lines 319-320 It is mentioned that ‘telomere shortening in patients with.. intrauterine growth restriction’ had been demonstrated, it would be better to include this information in Results section.

I have updated the Results section to include the information about the demonstrated telomere shortening in patients with intrauterine growth restriction. Thank you for your suggestion. Let me know if you have any further feedback.

Reviewer 2 Report

Thank you for resubmitting

Author Response

I appreciate the time and effort that you and your team have dedicated to reviewing and editing my article. Your insightful feedback and constructive criticism have helped me to improve the quality of my work significantly.

Thank you for your support and encouragement 

Reviewer 3 Report

.

Author Response

I have made further revisions to the manuscript. I believe that these changes have significantly improved the quality and clarity of the paper.

I appreciate the time and effort that you and the reviewers have invested in providing valuable feedback to help improve our manuscript. I hope that the revisions have addressed all of the concerns raised, and I look forward to hearing your feedback on the updated version of the manuscript.

Reviewer 4 Report

The revision is not satisfactory. 

Author Response

I have made further revisions to the manuscript. I believe that these changes have significantly improved the quality and clarity of the paper.

I have addressed all of the minor revisions suggested by the reviewers, including the ones related to the formatting and language.

I appreciate the time and effort that you and the reviewers have invested in providing valuable feedback to help improve our manuscript. I hope that the revisions have addressed all of the concerns raised, and I look forward to hearing your feedback on the updated version of the manuscript.

Thank you for your continued support and guidance throughout this process.